DATA RELEASE

# Distribution of flies of medical importance in Thailand: a dataset

Hassan Niyomdecha[1], Gerard Duvallet[2], Watthanasak Lertlumnaphakul[1], Ratchadawan Ngan-Klan[1,3], John Aerol Nobleza[1], Chauwat Charoenwiriyapap[4], Sylvie Manguin[5] and Theeraphap Chareonviriyaphap[1,3,*]

1 Department of Entomology, Faculty of Agriculture, Kasetsart University, Bangkok 10900, Thailand
2 Centre d'Écologie Fonctionnelle et Évolutive, Université Montpellier, CNRS, EPHE, IRD, Université Paul Valéry Montpellier 3, 34199 Montpellier, France
3 Research and Lifelong Learning Center for Urban and Environmental Entomology, Kasetsart University Institute for Advanced Studies, Kasetsart University, Bangkok 10900, Thailand
4 Department of Computer Engineering, Faculty of Engineering, Kasetsart University, Bangkok 10900, Thailand
5 HSM, Univ. Montpellier, CNRS, IRD, Montpellier 34090, France

## ABSTRACT

Non-mosquito Diptera of medical and veterinary importance, including both biting and non-biting species in the order *Diptera*, play a significant role in the transmission of disease pathogens, either as mechanical or biological vectors. In this review, a total of 3,492 records across seven families were identified, comprising 2,512 biting flies and 980 non-biting flies. Among the biting flies, the most frequently recorded family was *Simuliidae*, followed by *Calliphoridae*, *Muscidae*, *Psychodidae*, and *Tabanidae*. The majority of these records originated from northern Thailand and were previously published in various peer-reviewed journals.

**Subjects** Ecology, Biodiversity, Taxonomy

**Submitted:** 24 July 2025

\* Corresponding author. E-mail: faasthc@ku.ac.th

Preprint submitted at https://doi.org/10.5281/zenodo.16975936

Included in the series: ***Vectors of human disease*** (https://doi.org/10.46471/GIGABYTE_SERIES_0002)

## DATA DESCRIPTION

### Context

Flies are common insects found throughout Thailand, comprising both biting and non-biting species. Some of these species have significant implications for human and animal health, as they are capable of transmitting pathogens responsible for infectious diseases. Non-biting flies are known to process the mechanical transmission of pathogens via their body surfaces and are also considered nuisance pests. In contrast, biting flies can act as biological vectors, transmitting pathogens through their blood-feeding behavior. Understanding the spatial distribution of flies is essential for developing effective fly control strategies and reducing the risk of disease transmission.

This dataset, which documents the distribution of flies in Thailand from 1998 to 2023, includes a total of 3,492 records spanning seven identified families: Calliphoridae, Ceratopogonidae, Muscidae, Psychodidae, Sarcophagidae, Simuliidae, and Tabanidae. Each record comprises information across three primary categories: (i) Taxonomy – including scientific name, kingdom, phylum, class, order, family, genus, specific epithet, scientific name authorship, and taxon rank; (ii) Collection details – including event ID, occurrence ID, event date, and sampling protocol; (iii) Geolocation data – including county, country code, locality, location ID, decimal latitude, decimal longitude, and geodetic datum.

**Table 1.** Number of biting and non-biting flies collected by family.

| Type of flies | Family | Number of records |
|---|---|---|
| Biting flies | Ceratopogonidae | 77 |
| | Psychodidae | 204 |
| | Simuliidae | 1,824 |
| | Tabanidae | 191 |
| | Muscidae (biting) | 216 |
| Sub-total | | 2,512 |
| Non-biting flies | Calliphoridae | 711 |
| | Muscidae (non-biting) | 146 |
| | Sarcophagidae | 123 |
| Sub-total | | 980 |
| Total | | 3,492 |

**Table 2.** Geographic distribution of biting and non-biting flies collected by region.

| Region | Biting flies | Non-biting flies |
|---|---|---|
| Northern | 1,080 | 404 |
| Northeastern | 445 | 117 |
| Central | 219 | 234 |
| Eastern | 100 | 24 |
| Western | 245 | 92 |
| Southern | 278 | 38 |

An overview of the dataset obtained solely from the literature search is presented through three summary tables. Table 1 displays the total number of collected fly specimens in Thailand, classified by type – biting or non-biting – and organized by family. The most frequently recorded family is Simuliidae (*n* = 2,512), which accounts for 52.2% of all entries. Biting flies represent the majority, comprising 65.75% of all specimens. Within this group, Simuliidae is the most prevalent family (79.44%), while Calliphoridae is the most dominant among non-biting flies (58.70%). Table 2 illustrates the geographic distribution of biting and non-biting flies across six regions of Thailand. Notably, the Central region is the only area where non-biting flies outnumber biting flies. Table 3 summarizes the number of specimens collected using different sampling methods, and the manual collection by forceps yields the highest number of specimens overall.

## METHODS

### Literature search strategy

A literature search was conducted to identify studies on medically important biting and non-biting flies in Thailand published between 1998 and 2023. Six database platforms were used: Google Scholar, PubMed, Scopus, Global Index Medicus, the Cochrane Library, and the Kasetsart University Library website. The primary search terms included "Fly OR Flies" and "Thailand". Additional keywords to identify studies related to fly collection, surveillance, and distribution included "Trap OR Collection", "Spatial distribution", "Survey OR Surveillance", and environment types "Forest OR Coast", "Fruit Orchard OR Plantations", "Rural OR Urban", and "Community areas OR Farm areas". The selection of relevant studies was based on specific criteria. We included research articles that were published between 1998 and 2023, conducted within Thailand, and focused on biting or non-biting flies of medical importance. Additionally, the studies had to provide information on fly collection methods, habitats, or spatial distribution. On the other hand, excluded studies were those

**Table 3.** Number of flies collected by sampling method and family.

| Sampling method | Family | Number of records |
|---|---|---|
| Baited fly trap | Muscidae | 15 |
| CDC light trap | Simuliidae | 9 |
| | Psychodidae | 204 |
| | Ceratopogonidae | 7 |
| CDC-UV light trap | Ceratopogonidae | 17 |
| Combined manual methods[a] | Simuliidae | 33 |
| Funnel trap | Calliphoridae | 297 |
| | Muscidae | 70 |
| | Rhiniidae | 12 |
| | Sarcophagidae | 44 |
| Malaise trap | Simuliidae | 37 |
| | Tabanidae | 10 |
| Manual collection (forceps) | Ceratopogonidae | 53 |
| | Simuliidae | 1,388 |
| Nzi trap | Calliphoridae | 6 |
| | Muscidae | 21 |
| | Simuliidae | 5 |
| | Tabanidae | 181 |
| Sweep netting | Calliphoridae | 120 |
| | Muscidae | 38 |
| | Sarcophagidae | 52 |
| | Simuliidae | 188 |
| Vavoua trap | Muscidae | 188 |
| Miscellaneous/non-specific trap method[b] | Calliphoridae | 275 |
| | Muscidae | 41 |
| | Sarcophagidae | 27 |
| | Simuliidae | 161 |

[a]Combined manual methods, including human landing collection, sweep netting, and resting site collections.
[b]No data on the collection method was available in the original studies.

conducted outside Thailand, those that lacked sufficient methodological details, especially concerning sampling procedures, and any materials that had not been peer-reviewed, such as news articles or opinion reports. Relevant publications were reviewed, and data were extracted and summarized into categorized tables. The overall number of fly specimens collected by family from each study, with classification into biting and non-biting flies, is shown in Table 1, geographic distribution by region in Table 2, and sampling method by family in Table 3.

## Fly collections

The surveillance of both adult and immature stages of flies can provide informative and accurate data on the distribution and spatial movement of species within specific areas. This requires rigorous, longitudinal field sampling to identify habitats, harborages, breeding sites, and seasonal distribution, thereby enabling a deeper understanding of fly population dynamics. This review compiles studies conducted across Thailand on experimental research and population surveillance of medically and veterinary important biting and non-biting flies (excluding mosquitoes). Studies conducted between 1998 and 2023 that documented fly species from various families were included and summarized (Table 1). Specimens of adult flies were collected using standardized sampling protocols that combined both passive and active trapping methods. Passive methods included funnel traps, CDC light traps (standard and UV), Malaise traps, and baited traps using



approximately 300 g of raw chicken, pork, or squid. Nzi and Vavoua traps were employed in rural, farm, and forest-edge environments to target biting flies. These traps were deployed in diverse habitats such as residential areas, markets, farms, fruit orchards, forest edges, and coastal zones. Larvae and pupae were manually collected using forceps from moist substrates like leaf litter, stream margins, and soil (Table 3).

## Sampling protocol

Active collections, such as sweep nets, were conducted early morning (06:00–09:00 h) and late afternoon (16:00–18:00 h), when flies exhibit peak diurnal activity. Trap deployments typically lasted between 12 and 24 h, the exact duration depending on local weather and environmental conditions. Specimens were collected daily, immediately following trap retrieval. Immature fly specimens (larvae and pupae) were collected from natural aquatic and semi-aquatic habitats, such as submerged vegetation, wet soil substrates, muddy stream margins, rocks, and grasses along watercourses. Specimens were carefully collected using fine-tipped forceps to minimize damage. Preservation of adult and immature specimens was achieved by storing them in 95% ethanol, with capturing time, date, and location carefully logged for reference [1], fixing them in Carnoy's solution 3:1 ratio of 95% ethanol to glacial acetic acid and freezing at −20 °C pending further morphological and molecular analyses [2].

Subsequent morphological identification was conducted using established taxonomic keys relevant to each fly family studied. Selected voucher specimens were air-dried and pinned to serve as permanent reference collections. Identification of collected fly species relied primarily on morphological criteria using authoritative identification keys for each targeted family: Calliphoridae [3–5], Ceratopogonidae [6, 7], Muscidae [8–14], Psychodidae [15–26], Tabanidae [27, 28], Sarcophagidae [29, 30], and Simuliidae [31–58]. In addition, to resolve morphologically cryptic species complexes, we used molecular identification via DNA barcoding of mitochondrial cytochrome c oxidase subunit I (COI) and nuclear 18S/ITS1 gene regions [59–65]. Final species confirmations integrated both morphological characteristics and molecular data to ensure robust and accurate species determinations [66]. Besides morphological identification, molecular methods were also reported in the studies reviewed to improve species resolution within selected dipteran families. For Ceratopogonidae and Psychodidae, molecular identification was particularly crucial, as certain species exhibit indistinct or overlapping morphological traits, complicating accurate identification based on morphology alone [6, 20, 23]. Additionally, DNA barcoding targeting mitochondrial COI and nuclear 18S/ITS1 gene regions was used specifically for distinguishing closely related or cryptic species groups [59–63, 65–74]. Integrating molecular approaches alongside traditional morphological methods provided enhanced accuracy and robustness in species identification, as DNA barcoding techniques (PCR-based methods) have demonstrated high precision and reliability specifically for black fly taxonomy [2].

## Data georeferencing process

The decimal degree (DD) system and World Geodetic System 1984 (WGS84) were used to determine the geographic coordinates (latitude and longitude) of given location in dataset before generating a Geographic Information System (GIS) map of flies of medical and veterinary significance.

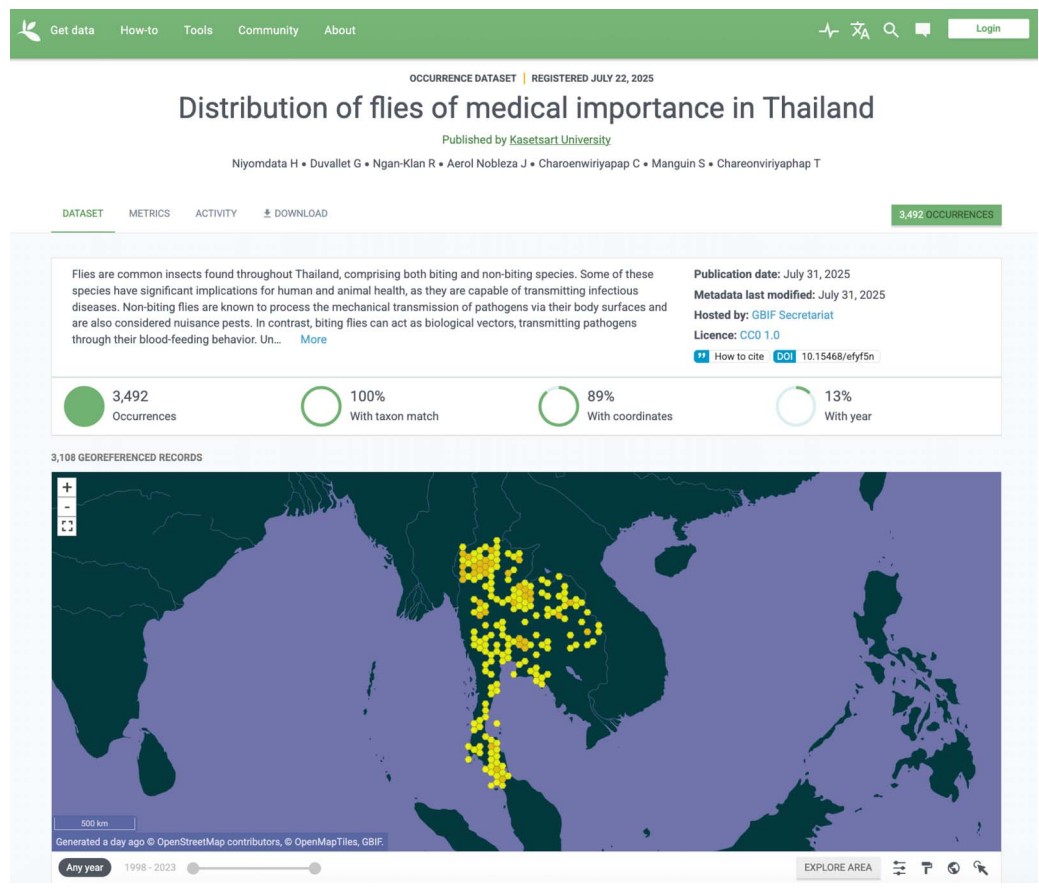

**Figure 1.** Interactive map of the georeferenced occurrences hosted by GBIF [75]. https://www.gbif.org/dataset/0c0ab16f-7565-4b64-9e6d-939d82f4d144

## DATA VALIDATION AND QUALITY CONTROL

Fly identification was performed by the respective article authors and cleaned by Prof. Gérard Duvallet, a world-renowned fly specialist, as documented in the literature. The dataset is presented in Darwin Core format and shared via the Global Biodiversity Information Facility (GBIF). All required fields are present and have undergone screening in GBIF's Integrated Publishing Toolkit (IPT). Metadata is also provided with the published resource in GBIF [75].

## REUSE POTENTIAL

These data are critical because they provide records for the distribution of medically important flies collected at different sites in Thailand (Figure 1). The fly biodiversity in the northern region may be related to variations in climate, geography, and elevation. The data can be reused by academia, government, civil society, and non-governmental organizations to improve surveillance and control activities.

## DATA AVAILABILITY

The data supporting this article are published through the GBIF Asia IPT and are available under a CC0 license from GBIF [75].

## EDITORS' NOTE

This paper is part of a series of Data Release articles working with GBIF and supported by TDR, the Special Programme for Research and Training in Tropical Diseases hosted at the World Health Organization, in order to publish datasets on vectors of human diseases [76].

## LIST OF ABBREVIATIONS

COI, cytochrome c oxidase subunit I; GBIF, Global Biodiversity Information Facility; DD, decimal degree system; WGS84, World Geodetic System 1984; GIS, Geographic Information System; IPT: Integrated Publishing Toolkit.

## DECLARATIONS

### Ethical approval and consent to participate

The authors declare that ethical approval was not required for this type of research.

### Competing interests

The author(s) declare that they have no competing interests.

### Authors' contributions

Conceptualization: HN, WL, JAN, RN-G, GD, SM, TC. Data curation: HN, WL, JAN, RN-G, CC, TC. Methodology: HN, WL, JAN, RN-G, GD, TC. Writing – original draft preparation: HN, WL, JAN, CC, SM, TC. Writing – review and editing: HN, WL, JAN, CC, GD, SM, TC. Supervision: TC, SM. Funding acquisition: TC.

### Funding

This study was funded by the Kasetsart University Research and Development Institute (KURDI) Fundamental Fund program [FF (KU) 14.68].

### Acknowledgements

The authors would like to thank everybody who contributed to the creation of these datasets and papers. Special gratitude to Paloma Helena Fernandes Shimabukuro for her suggestions on the dataset and manuscript preparation.

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
