## [Editor Report]

Editor’s AssessmentThis paper is a submission to the GigaByte vectors of human disease series, and shares observation data of non-mosquito flies of medical and veterinary importance across Thailand. This includes both biting and non-biting species in the order Diptera which play a significant role in the transmission of disease pathogens, either as mechanical (via their body surfaces) or biological (transmitting through their feeding) vectors. This data is curated from the published literature (published between 1998-2023) a total of 3,492 records across seven families were identified, comprising 2,512 biting flies and 980 non-biting flies and shared via the Global Biodiversity Information Facility (GBIF). Peer review and data audits were carried out to validate the quality of this data. Demonstrating it should be extremely useful for reuse by academia, government, civil society, and non-governmental organizations to improve surveillance and control activities.Editor’s AssessmentThis paper is a submission to the GigaByte vectors of human disease series, and shares observation data of non-mosquito flies of medical and veterinary importance across Thailand. This includes both biting and non-biting species in the order Diptera which play a significant role in the transmission of disease pathogens, either as mechanical (via their body surfaces) or biological (transmitting through their feeding) vectors. This data is curated from the published literature (published between 1998-2023) a total of 3,492 records across seven families were identified, comprising 2,512 biting flies and 980 non-biting flies and shared via the Global Biodiversity Information Facility (GBIF). Peer review and data audits were carried out to validate the quality of this data. Demonstrating it should be extremely useful for reuse by academia, government, civil society, and non-governmental organizations to improve surveillance and control activities.

---

## [Reviewer Report]

Upload additional filesDRR-202507-03-R01/stage_files/DRR-202507-03/Review MS/DRR-202507-03_Data-Review-BM.docx.pdfReviewer name and names of any other individual's who aided in reviewer Bastien MolcretteDo you understand and agree to our policy of having open and named reviews, and having your review included with the published papers. (If no, please inform the editor that you cannot review this manuscript.)YesIs the language of sufficient quality?YesPlease add additional comments on language quality to clarify if needed
Are all data available and do they match the descriptions in the paper? YesAdditional Commentssome differences observed between the occurrences in Tables 1 & 3, and the GBIF dataset (see attached report)Are the data and metadata consistent with relevant minimum information or reporting standards? See GigaDB checklists for examples <a href="http://gigadb.org/site/guide" target="_blank">http://gigadb.org/site/guide</a>YesAdditional CommentsIs the data acquisition clear, complete and methodologically sound?YesAdditional CommentsIs there sufficient detail in the methods and data-processing steps to allow reproduction?YesAdditional CommentsIs there sufficient data validation and statistical analyses of data quality? YesAdditional CommentsIs the validation suitable for this type of data?YesAdditional CommentsIs there sufficient information for others to reuse this dataset or integrate it with other data?YesAdditional CommentsAny Additional Overall Comments to the AuthorChange GBIF dataset license to CC0 public domainRecommendationMinor Revision

---

## [Reviewer Report]

Upload additional filesDRR-202507-03-R01/stage_files/DRR-202507-03/Review MS/Reviewer_Report_for_Manuscript_DRR.docxReviewer name and names of any other individual's who aided in reviewer Dr Zubaidah Ya'cobDo you understand and agree to our policy of having open and named reviews, and having your review included with the published papers. (If no, please inform the editor that you cannot review this manuscript.)YesIs the language of sufficient quality?YesPlease add additional comments on language quality to clarify if needed
Are all data available and do they match the descriptions in the paper? NoAdditional CommentsThe abstract mentions "nine families," but only seven are listed in the Methods section (Calliphoridae, Ceratopogonidae, Muscidae, Psychodidae, Sarcophagidae, Simuliidae, Tabanidae). Please clarify this discrepancy. Therefore, I suggest listing all nine families explicitly or correcting the abstract.Are the data and metadata consistent with relevant minimum information or reporting standards? See GigaDB checklists for examples <a href="http://gigadb.org/site/guide" target="_blank">http://gigadb.org/site/guide</a>YesAdditional CommentsThe manuscript aligns with GigaByte’s scope for concise, data-focused publications and meets GBIF’s quality standards. Only pending resolution of major issues.Is the data acquisition clear, complete and methodologically sound?NoAdditional CommentsMajor issues: -The abstract mentions "nine families," but only seven are listed in the Methods section (Calliphoridae, Ceratopogonidae, Muscidae, Psychodidae, Sarcophagidae, Simuliidae, Tabanidae). Please clarify this discrepancy. Therefore, I suggest listing all nine families explicitly or correcting the abstract. -Northern Thailand dominates the records (Table 2), potentially skewing conclusions about nationwide distribution. I suggest acknowledging this bias and discussing its implications for generalizability. -The "combined manual methods" and "miscellaneous/non-specific trap methods" (Table 3) lack detail. Suggest providing brief descriptions (e.g., "human landing collections") or citing protocols. -Only a minor issue on the formatting, I found inconsistent spacing/typos (e.g., "Distribution.f / iteso.fm edical.mportance.nz-hailand.aataset.pdate_uth.DOI.docx" in headers). So, it is better to clean up auto-generated artifacts for professionalism. Some citations are redundant (e.g., multiple entries for Simuliidae). Authors please consolidate or use "et al." for sequential works by the same authors. -In the abstract, the phrase "flies of medical and veterinary importance (excluding mosquitoes)" could be streamlined (e.g., "non-mosquito Diptera of medical/veterinary importance").Is there sufficient detail in the methods and data-processing steps to allow reproduction?NoAdditional CommentsSee comments for major issues.Is there sufficient data validation and statistical analyses of data quality? YesAdditional CommentsSee comments for major issues.Is the validation suitable for this type of data?NoAdditional CommentsSee comments for major issues.Is there sufficient information for others to reuse this dataset or integrate it with other data?YesAdditional CommentsThis manuscript will benefit the scientific community, particularly in the field of infectious disease transmission. I found that the dataset is well-positioned for reuse in modelling disease transmission risks, comparative studies across Southeast Asia and monitoring climate change impacts on fly distributions.Any Additional Overall Comments to the AuthorThe manuscript aligns with GigaByte’s scope for concise, data-focused publications and meets GBIF’s quality standards. However, pending resolution of major issues.RecommendationMinor Revision